# The Risk Factors for Development of Type 2 Diabetes: Panasonic Cohort Study 4

**DOI:** 10.3390/ijerph19010571

**Published:** 2022-01-05

**Authors:** Yuto Saijo, Hiroshi Okada, Masahide Hamaguchi, Momoko Habu, Kazushiro Kurogi, Hiroaki Murata, Masato Ito, Michiaki Fukui

**Affiliations:** 1Department of Endocrinology and Metabolism, Graduate School of Medical Science, Kyoto Prefectural University of Medicine, 465 Kajii-cho, Kawaramachi-Hirokoji, Kyoto 602-8566, Japan; y-saijo@koto.kpu-m.ac.jp (Y.S.); mhama@koto.kpu-m.ac.jp (M.H.); michiaki@koto.kpu-m.ac.jp (M.F.); 2Department of Diabetes and Endocrinology, Matsushita Memorial Hospital, Moriguchi 570-8540, Japan; habu.momoko@jp.panasonic.com; 3Department of Health Care Center, Panasonic Health Insurance Organization, Moriguchi 570-8540, Japan; kurogi.kazushirou@jp.panasonic.com (K.K.); itoh.masato@jp.panasonic.com (M.I.); 4Department of Orthopaedic Surgery, Matsushita Memorial Hospital, Moriguchi 570-8540, Japan; murata.hiroaki@jp.panasonic.com

**Keywords:** type 2 diabetes, body mass index, database research, cohort study

## Abstract

(1) Background: As no study has evaluated the risk factors contributing to the development of type 2 diabetes according to body weight mass (BMI) categories in a large cohort and over a long period in a Japanese population, we aimed to assess them in 46,001 Japanese individuals; (2) Methods: This long-term retrospective cohort study included individuals who participated in a medical health checkup program at Panasonic Corporation from 2008 to 2018. In total, 3,208 participants developed type 2 diabetes. The BMI at 2008 were divided into 3 groups: BMI < 22 kg/m^2^, 22 ≤ BMI < 25 kg/m^2^, and BMI ≥ 25 kg/m^2^ to evaluate the risk factors for type 2 diabetes; (3) Results: Cox regression analyses revealed that the significant risk factors were age, BMI, systolic blood pressure, low-density lipoprotein cholesterol, high-density lipoprotein cholesterol, triglycerides, fasting plasma glucose, current smoking status, and alcohol consumption in populations with BMI ≥ 25.0 kg/m^2^. The risk factors in the population with 22 ≤ BMI < 25 kg/m^2^ were identical to that of BMI ≥ 25.0 kg/m^2^ excluding systolic blood pressure, low-density lipoprotein cholesterol, and skipping breakfast. However, in the population with BMI < 22.0 kg/m^2^, no association was found as to BMI, SBP, LDL cholesterol level, and triglycerides; (4) Conclusions: The risk factors for developing diabetes were quite different between the population with BMI < 22 kg/m^2^ and the population with 22 ≤ BMI < 25 kg/m^2^ or BMI ≥ 25 kg/m^2^ in a Japanese population.

## 1. Introduction

In recent decades, diabetes has become increasingly prevalent globally, causing the health expenditure to pile up. Therefore, preventing the development of diabetes is of great significance and a common concern in clinical settings. Previous studies indicate that multiple risk factors including age, family history, obesity [1], impaired fasting glycemia [2,3], impaired glucose tolerance [2], nonalcoholic fatty liver disease [4], uric acid [5], potassium [6], low lung function [7], smoking status, hypertension, hyperlipidemia, chronic inflammation [8,9,10], liver enzymes [11,12], lifetime maximum BMI, change in BMI from early adulthood [13], eating speed [14], and lack of sleep [15] accelerate the development of diabetes in the Japanese population. We should identify those who are at a high risk of developing type 2 diabetes. Health guidance for prevention of diabetes is necessary at an early stage of prediabetes. 

There are fewer obese people in Japan than in Western countries. BMIs of Japanese patients with diabetes are similar to those of the general Japanese population [16]. Unlike Western populations, pre-diabetes in the Japanese population is characterized by both relatively low insulin secretion and insulin resistance. [17]. It has been reported that Asian subjects developed diabetes at lower ranges of BMI than those of Western populations [18]. In other words, type 2 diabetes is a heterogeneous disease caused by multifactorial etiology in the Japanese population. Because the etiology of diabetes in the Japanese population is characterized by lower insulin secretion and insulin resistance, we thought that diminished insulin secretion might have a direct impact on developing diabetes especially in a lean or non-obese population. Therefore, we believe that the risk factors for the development of diabetes might differ according to BMI categories and its appropriateness among BMI categories has been questioned. We thought that the risk factors for the development of diabetes needed to be evaluated separately according to BMI categories in a Japanese population.

As far as we know, no other study has elucidated the risk factors contributing to the development of type 2 diabetes according to BMI categories in a large cohort and a long period in a Japanese population. Therefore, this study, for the first time, looked into the risk factors for type 2 diabetes in a Japanese population according to BMI categories.

## 2. Materials and Methods

### 2.1. Study Design and Data Collection

This is a long-term retrospective cohort study. The subjects consist of corporate employees who received a medical health checkup program conducted in Panasonic Corporation, Osaka, Japan. This program aims to promote public health by detecting as early as possible the signs of chronic diseases including metabolic disorders and evaluating what risk factors underlie them. Every employee participates in this program every year. The employers are required by the Ordinance of the Ministry of Health, Labor and Welfare, to provide employees with medical examinations conducted by a physician. We obtained the data between 2008 and 2018 from the database which we call the Panasonic cohort study.

The blood tests were performed after more than 10 hours of fasting. Height and body weight were recorded using an instrument capable of measuring both automatically. The change in BMI was defined as the delta BMI (BMI in the end of study, which means the development of diabetes or 2018—BMI in 2008). We used the standardized self-administered questionnaire, which has been previously validated, to assess the baseline characteristics in 2008. Smoking status was categorized into current smokers, past smokers, and non-smokers. Eating speed was classified as fast, normal, and slow. We asked participants if they made a habit of eating breakfast or a snack after dinner. Alcohol drinkers were defined by daily consumption of alcohol. Regular exercisers were defined as those who played any sport twice a week regularly. The definition of Type 2 diabetes in this study was as follows: fasting plasma glucose (FPG) level ≥126 mg/dL, having a self-reported history of diabetes, and/or the administration of anti-diabetic agents.

This study was approved by the local ethics committee of Panasonic Health Insurance Organization (Approval number: 2021-001) and was conducted in accordance with the principles of the Declaration of Helsinki.

### 2.2. Exclusion Criteria

A total of 140,590 employees received a medical health checkup in 2008. The participants who did not consecutively receive health checkups from 2008 to 2018 were excluded (*n* = 65,256) from the study. Those with diabetes at baseline were excluded (*n* = 4342). Those with missing data were also excluded (*n* = 24,991). 

### 2.3. Statistical Analyses

We calculated frequencies and the mean values of potential confounding variables. The differences in general characteristics at baseline according to BMI categories were assessed by analysis of variance and chi-square test as appropriate. The BMI were divided into 3 groups: BMI < 22 kg/m^2^, 22 ≤ BMI < 25 kg/m^2^, and BMI ≥ 25 kg/m^2^ because Japan society for the study of obesity defined 22 kg/m^2^ of BMI as standard BMI, which is associated with the lowest morbidity and defined BMI ≥ 25 kg/m^2^ as obesity [19]. The hazard ratios for the development of type 2 diabetes were calculated by Cox regression analyses using multivariate models. The factors related to the development of type 2 diabetes are selected as covariates included in the multivariate model. The multivariate model was adjusted for age, sex, BMI, systolic blood pressure (SBP), low-density lipoprotein (LDL) cholesterol, high-density lipoprotein (HDL) cholesterol, triglycerides, uric acid, smoking status, eating speed, snack after dinner, skipping breakfast, alcohol consumption, and physical exercise. We adopted multivariate logistic regression analysis to demonstrate the effects of the change in BMI from 2008 to 2018 on the development of type 2 diabetes in 2018. The covariates in the multiple logistic regression model were all variables in Cox regression analyses plus the change in BMI from 2008 to 2018. The hazard ratios were assessed for the development of type 2 diabetes in patients with BMI < 22 kg/m^2^, 22.0 ≤ BMI < 25.0 kg/m^2^, and BMI ≥ 25.0 kg/m^2^ at baseline. All continuous variables are presented as mean ± SD or absolute number. The differences with *p* value < 0.05 are regarded as statistically significant. The associations are presented as hazard ratios with a 95% confidence interval (CI). We used JMP software, version 10 (SAS Institute, Cary, NC, USA) to perform statistical analyses.

## 3. Results

Table 1 shows the baseline characteristics of the participants enrolled in this study. In total, 3208 participants among all participants developed type 2 diabetes during the study period. Table 1 also shows the differences in characteristics at baseline according to BMI categories.

Table 2 shows the unadjusted hazards ratios for the development of diabetes. Table 3 shows the adjusted hazards ratios in multivariate models for the development of diabetes. The significant risk factors for developing diabetes were age, HDL cholesterol, FPG, smoking status, and eating speed in the population with BMI < 22.0 kg/m^2^. The significant risk factors for developing diabetes were age, sex, BMI, HDL cholesterol, triglycerides, FPG, smoking status, skipping breakfast, and alcohol consumption in the population with 22.0 ≤ BMI < 25.0 kg/m^2^. The significant risk factors for developing diabetes were age, BMI, SBP, LDL cholesterol, HDL cholesterol, triglycerides, FPG, smoking status, and alcohol consumption in the population with BMI ≥ 25.0 kg/m^2^. If we put SBP, LDL cholesterol, HDL cholesterol, triglycerides, and FPG dichotomized into multivariate models, we found that skipping breakfast was significantly associated with developing diabetes in the population with BMI < 22.0 kg/m^2^ (HR; 1.29, 95% CI; 1.02–1.61, *p* = 0.03), with 22.0 ≤ BMI < 25.0 kg/m^2^ (HR; 1.29, 95% CI; 1.11–1.49, *p* = 0.0009), and with BMI ≥ 25.0 kg/m^2^ (HR; 1.14, 95% CI; 1.03–1.28, *p* = 0.02).

The change in BMI was higher in the population without developing diabetes than that with developing diabetes in the population with BMI < 22.0 kg/m^2^ (0.89 ± 1.44 vs. 0.61 ± 1.42 kg/m^2^; *p* < 0.0001). In contrast, the change in BMI was higher in the population with developing diabetes than that without developing diabetes in the population with 22.0 ≤ BMI < 25.0 kg/m^2^ and with BMI ≥ 25.0 kg/m^2^ (0.67 ± 1.54 vs. 0.54 ± 1.61 kg/m^2^; *p* = 0.01 and 0.73 ± 1.77 vs. 0.31 ± 2.17 kg/m^2^; *p* < 0.0001, respectively). In other words, the rate of weight gain is positively correlated with the development of type 2 diabetes in the population with 22.0 ≤ BMI < 25.0 kg/m^2^ and with BMI ≥ 25.0 kg/m^2^. However, it is negatively correlated with the development of type 2 diabetes in the population with BMI < 22.0 kg/m^2^. The multivariate logistic regression analysis revealed that the change in BMI from 2008 to 2018 was associated in the population with 22.0 ≤ BMI < 25.0 kg/m^2^ (odds ratio; 1.13, 95% CI; 1.07–1.19, *p* < 0.0001) and with BMI ≥ 25.0 kg/m^2^ (odds ratio; 1.09, 95% CI; 1.01–1.12, *p* < 0.0001). However, the change in BMI from 2008 to 2018 was not associated with the development of type 2 diabetes (odds ratio; 1.04, 95% CI; 0.94–1.16, *p* = 0.41). 

## 4. Discussion

The significant risk factors for the developing diabetes were different among BMI categories. In the population with BMI ≥ 25.0 kg/m^2^, age, obesity, hypertension, hyperlipidemia, FPG level, and smoking status were associated with the development of type 2 diabetes, which have been well known as risk factors for developing diabetes. The significant risk factors for developing diabetes in the population with 22.0 ≤ BMI < 25.0 kg/m^2^ were almost identical to those of BMI ≥ 25.0 kg/m^2^. In contrast, in population with BMI < 22.0 kg/m^2^, BMI, SBP, LDL cholesterol level and triglycerides were not associated with the development of type 2 diabetes. Instead, an association with age, HDL cholesterol level, FPG level, smoking status and eating speed were observed in this population. The major finding of this study is that the risk factors for developing diabetes significantly vary between the population with BMI < 22.0 kg/m^2^ and the population with 22.0 ≤ BMI < 25.0 kg/m^2^ or BMI ≥ 25.0 kg/m^2^ in a Japanese population.

Type 2 diabetes is well known to have heterogeneous etiology in the Japanese population. The difference in risk factors between each BMI category implies that they are different in etiology. It has been well established that type 2 diabetes originates from relative insulin deficiency and/or insulin resistance. There are some studies indicating that obesity induces chronic inflammation [20] and insulin resistance [21,22]. Insulin resistance is partly attributed to the dysregulation of adipocytes, which produce and secrete several cytokines called adipocytokines, e.g., tumor necrosis factor-α(TNF-α), adiponectin, leptin, and plasminogen activator inhibitor-1 (PAI-1) [23,24]. TNF-α develops insulin resistance, whereas adiponectin improves insulin sensitivity. Weight gain is associated with increased visceral adipose tissue, causing the adipocytes to produce more TNF-α and less adiponectin. As a result, higher BMI contributes to insulin resistance and subsequent relative insulin deficiency, culminating in hyperglycemia. These factors play a key role in metabolic syndrome, which refers to a set of metabolic disruptions including visceral obesity, hyperglycemia, dyslipidemia, and hypertension, which are closely related to each other [25]. In our study, dyslipidemia and hypertension were also associated with developing diabetes in the population with BMI ≥ 25.0 kg/m^2^. Thus, as for population with BMI ≥ 25.0 kg/m^2^, because they are obese, they are more likely to have higher insulin resistance compared to the population with BMI < 22, and once a chronic inflammatory state has been created, these metabolic disruptions fall into a vicious cycle. Interestingly, the same applies to the population with 22.0 ≤ BMI < 25.0 kg/m^2^, although they are within the range of normal weight. This result suggests that weight gain and a metabolic disorder in patients within a normal weight range can raise the possibility of developing diabetes. In contrast, for the population with BMI < 22, dyslipidemia, hypertension, or even body weight are not significant risk factors for diabetes. Hyperglycemia in the absence of obesity probably means diminished insulin secretion. Diminished insulin secretion is characteristic of patients with diabetes in Asian countries [26], where BMIs of patients are not remarkably greater than those of the general population [16]. They are possibly more susceptible to reduced insulin secretion and aging than the other populations. Interestingly, the change in BMI was higher in the population without developing diabetes than that with developing diabetes in the population with BMI < 22.0 kg/m^2^ in this study, which might mean diminished insulin secretion. Smoking is one of the risk factors for type 2 diabetes, as proven by previous studies [27]. Our study reinforces the evidence by showing that regardless of BMI, smoking is a strong risk factor for diabetes.

The westernization of eating habits started after World War Ⅱ in Japan. Therefore, we suppose that eating habits are one of the most important factors causing higher BMI and developing diabetes in the Japanese population. In terms of eating habits, our study showed that eating speed and alcohol consumption had a protective effect in the population with BMI < 22.0 kg/m^2^ and with 22.0 ≤ BMI < 25.0 kg/m^2^ or with BMI ≥ 25.0 kg/m^2^, respectively. There are some studies reporting that eating fast is associated with obesity [28]. People who eat fast tend to consume more food before they achieve satiety, resulting in the increase in total energy intake. Moreover, eating fast is reported to adversely affect the postprandial glucose excursion [29], which would lead to the impairment of insulin secretion. Especially in lean or non-obese populations (BMI < 22.0 kg/ m^2^), insulin secretion might be relatively low compared to that of other populations. This is probably why eating slow showed a protective effect on developing type 2 diabetes in the population of BMIs < 22.0 kg/m^2^ in our study. As for alcohol consumption, there is no consistent evidence that alcohol itself has a protective effect on developing diabetes. How alcohol consumption influences the risk of type 2 diabetes might depend on the dose or the type of alcohol [30]. High alcohol consumption causes liver dysfunction, ultimately liver cirrhosis, which increases the risk of type 2 diabetes. Red wine contains antioxidants which might have a protective effect on developing diabetes. Another hypothesis is that alcohol drinkers consume a less balanced diet than non-drinkers, possibly leading to less energy intake.

In Japan, a nationwide screening program was introduced to identify the subjects with obesity (BMI ≥ 25.0 kg/m^2^) and risk factors of cardiovascular disease and to provide health guidance to prevent the development and progression of metabolic disorders and cardiovascular disease among the working-age population in 2008 [31]. However, Fukuma S. et al. have reported that the national health guidance intervention was not associated with the reduction of cardiovascular risk such as diabetes [32]. Further research is needed to specify the subjects at high risk of metabolic disorders with or without obesity. Therefore, we believe that our study, which evaluated the risk factors according to BMI categories, is meaningful and will contribute to understanding the specific method of lifestyle interventions for improving metabolic disorders.

The strength of our study is its long period, large population, and consecutive data. However, there are several limitations. First, the diagnosis of diabetes was not based on hemoglobin A1c level. However, a previous study revealed that the correlation coefficient between FPG level and hemoglobin A1c level was 0.85 [33]. Thus, we considered it acceptable to substitute FPG levels and a self-reported history of diabetes for hemoglobin A1c level in order to assess the development of diabetes. Second, we might not evaluate the exact date of development of type 2 diabetes and exact information of a history of diabetes because the subjects received a medical health checkup program once a year and we used the self-administered questionnaire to survey the history of diabetes. Third, it is well known that family history contributed to developing diabetes. Environmental factors including levels or types of incomes might have contributed to developing diabetes. Unfortunately, however, we had no data about family history of diabetes and details of levels or types of incomes. Forth, the participants consisted of relatively young Japanese people and this cohort was a selected working age cohort, which was composed of a single corporation. Therefore, it might not be obvious whether we can generalize what we found in this study to other generations and ethnic groups. We need to consider future studies to generalize our findings. We think possible studies, which follow up 10 years in the future and are composed of several companies, are necessary. 

## 5. Conclusions

In clinical settings, it is crucial for us to perceive that the risk factors for diabetes differ among BMI categories in the Japanese population because it enables us to give advice tailored to each individual’s BMI, which might help prevent the development of diabetes. In conclusion, the risk factors for the development of type 2 diabetes are different between the population with BMI < 22.0 kg/m^2^ and the population with 22.0 ≤ BMI < 25.0 kg/m^2^ or BMI ≥ 25.0 kg/m^2^ in the Japanese population.

## Figures and Tables

**Table 1 ijerph-19-00571-t001:** Characteristics of participants.

	ALL	BMI < 22.0 kg/m^2^	22.0 ≤ BMI < 25.0 kg/m^2^	BMI ≥ 25.0 kg/m^2^	*p* Value
N	46,001	17,857	16,629	11,515	-
Age (y)	42.06 (6.09)	41.16 (6.44)	42.61 (5.83) *	42.67 (5.72) *	<0.0001
Sex (male/female)	38,569/7432	12,935/4922	15,092/1537	10,542/973	<0.0001
Body mass index (kg/m^2^)	23.09 (3.28)	20.05 (1.38)	23.36 (0.85) *	27.43 (2.41) *,#	<0.0001
SBP (mmHg)	118.45 (13.97)	113.48 (12.95)	119.09 (12.97) *	125.25 (13.83) *,#	<0.0001
DBP (mmHg)	74.11 (10.67)	70.44 (9.88)	74.64 (10.08) *	79.03 (10.56) *,#	<0.0001
LDL cholesterol (mg/dL)	123.91 (30.86)	113.52 (28.93)	127.17 (29.81) *	135.29 (30.18) *,#	<0.0001
HDL cholesterol (mg/dL)	59.20 (14.81)	65.71 (15.15)	57.42 (13.42) *	51.66 (11.54) *,#	<0.0001
Triglycerides (mg/dL)	114.41 (88.40)	84.86 (62.17)	118.69 (84.55) *	154.04 (109.31) *,#	<0.0001
FPG (mg/dl)	92.61 (8.75)	90.40 (8.10)	93.11 (8.48) *	95.32 (9.21) *,#	<0.0001
Uric acid (mg/dL)	5.89 (1.38)	5.34 (1.29)	6.06 (1.27) *	6.50 (1.33) *,#	<0.0001
Smoking (none/past/current)	23,613/5910/16,478	10,019/1872/5966	8113/2410/6106	5481/1628/4406	<0.0001
Eating speed (fast/normal/slow)	15,838/27,032/3131	4304/11,691/1862	5975/9759/895	5559/5582/374	<0.0001
Snack after dinner (+/−)	7991/38,010	3023/14,834	2797/13,832	2171/9344	<0.0001
Skipping breakfast (+/−)	10,566/35,435	4035/13,822	3779/12,850	2752/8763	0.02
Alcohol drinker (+/−)	11,085/34,916	4228/13,629	4502/12,127	2355/9160	<0.0001
Physical exercise (+/−)	7009/38,992	2504/15,353	2741/13,888	1764/9751	<0.0001

Data are presented as mean (standard deviation) or absolute number. SBP, systolic blood pressure; DBP, diastolic blood pressure; LDL, low-density lipoprotein; HDL, high-density lipoprotein; FPG, fasting plasma glucose. *: *p* < 0.05 vs. BMI < 22.0 kg/m^2^, #: *p* < 0.05 vs. 22.0 ≤ BMI < 25.0 kg/m^2^.

**Table 2 ijerph-19-00571-t002:** Unadjusted hazard ratios for development of diabetes.

	BMI < 22.0 kg/m^2^		22.0 kg/m^2^ ≤ BMI < 25.0 kg/m^2^		BMI ≥ 25.0 kg/m^2^	
	HR (95% CI)	*p* Value	HR (95% CI)	*p* Value	HR (95% CI)	*p* Value
Age (per 1 year)	1.10 (1.08–1.12)	<0.0001	1.08 (1.07–1.10)	<0.0001	1.04 (1.03–1.05)	<0.0001
Sex (male)	3.28 (2.44–4.51)	<0.0001	2.12 (1.60–2.89)	<0.0001	1.23 (1.03–1.49)	0.02
Body mass index at baseline(per 1 kg/m^2^)	1.22 (1.13–1.32)	<0.0001	1.27 (1.18–1.36)	<0.0001	1.18 (1.16–1.19)	<0.0001
Systolic blood pressure(per 1 mmHg)	1.04 (1.03–1.04)	<0.0001	1.02 (1.01–1.02)	<0.0001	1.02 (1.02–1.03)	<0.0001
LDL cholesterol(per 1 mg/dL)	1.01 (1.007–1.01)	<0.0001	1.007 (1.005–1.009)	<0.0001	1.006 (1.004–1.007)	<0.0001
HDL cholesterol (per 1 mg/dL)	0.98 (0.98–0.99)	<0.0001	0.98 (0.98–0.99)	<0.0001	0.98 (0.97–0.98)	<0.0001
Triglycerides (per 1 mg/dL)	1.003 (1.002–1.004)	<0.0001	1.002 (1.0016–1.002)	<0.0001	1.001 (1.001–1.002)	<0.0001
Fasting plasma glucose (per 1 mg/dL)	1.17 (1.16–1.18)	<0.0001	1.16 (1.15–1.16)	<0.0001	1.13 (1.12–1.13)	<0.0001
Uric acid (per 1 mg/dL)	1.30 (1.21–1.40)	<0.0001	1.18 (1.12–1.23)	<0.0001	1.10 (1.06–1.14)	<0.0001
Smoking (past) (ref: none)	1.35 (0.96–1.85)	0.08	1.33 (1.09–1.61)	0.004	1.00 (0.86–1.16)	0.99
Smoking (current) (ref: none)	2.03 (1.66–2.47)	<0.0001	1.81 (1.59–2.08)	<0.0001	1.34 (1.22–1.48)	<0.0001
Eating speed (slow) (ref: normal)	0.56 (0.37–0.81)	0.002	0.86 (0.62–1.15)	0.31	0.87 (0.64–1.16)	0.35
Eating speed (fast) (ref: normal)	0.9999 (0.80–1.24)	0.99	1.12 (0.98–1.27)	0.09	1.18 (1.08–1.30)	0.0005
Snack after dinner (yes) (ref: no)	0.84 (0.64–1.08)	0.18	0.90 (0.76–1.06)	0.44	1.01 (0.89–1.13)	0.90
Skipping breakfast (yes) (ref: no)	1.34 (1.09–1.65)	0.007	1.33 (1.16–1.52)	<0.0001	1.23 (1.11–1.37)	<0.0001
Alcohol drinker (yes) (ref: no)	1.65 (1.35–2.01)	<0.0001	1.10 (0.96–1.26)	0.17	0.80 (0.70–0.90)	0.0002
Physical exercise (yes) (ref: no)	1.21 (0.93–1.55)	0.15	1.11 (0.95–1.30)	0.19	0.96 (0.84–1.09)	0.52

LDL, low-density lipoprotein; HDL, high-density lipoprotein.

**Table 3 ijerph-19-00571-t003:** Multivariate adjusted hazard ratios for development of diabetes.

	BMI < 22.0 kg/m^2^		22.0 kg/m^2^ ≤ BMI < 25.0 kg/m^2^		BMI ≥ 25.0 kg/m^2^	
	HR (95% CI)	*p* Value	HR (95% CI)	*p* Value	HR (95% CI)	*p* Value
Age (per 1 year)	1.06 (1.04–1.08)	<0.0001	1.04 (1.02–1.05)	<0.0001	1.02 (1.007–1.03)	0.001
Sex (male)	1.05 (0.72–1.54)	0.81	0.68 (0.49–0.96)	0.03	0.90 (0.73–1.11)	0.32
BMI at baseline (per 1 kg/m^2^)	1.03 (0.95–1.12)	0.45	1.11 (1.03–1.20)	0.008	1.10 (1.09–1.12)	<0.0001
Systolic blood pressure (per 1 mmHg)	1.005 (0.998–1.01)	0.14	0.9996 (0.995–1.004)	0.88	1.006 (1.002–1.009)	0.001
LDL cholesterol (per 1 mg/dL)	1.003 (0.999–1.006)	0.10	1.002 (0.99998–1.004)	0.05	1.003 (1.002–1.005)	0.0001
HDL cholesterol (per 1 mg/dL)	0.99 (0.98–0.998)	0.01	0.99 (0.99–0.998)	0.008	0.99 (0.98–0.99)	<0.0001
Triglycerides (per 1 mg/dL)	1.0005 (0.999–1.002)	0.44	1.0008 (1.0003–1.001)	0.0005	1.0005 (1.00008–1.0009)	0.02
Fasting plasma glucose (per 1 mg/dL)	1.16 (1.15–1.17)	<0.0001	1.16 (1.15–1.16)	<0.0001	1.12 (1.11–1.13)	<0.0001
Uric acid (per 1 mg/dL)	0.99 (0.91–1.07)	0.75	1.05 (0.997–1.11)	0.06	0.98 (0.94–1.01)	0.23
Smoking (past) (ref: none)	0.89 (0.63–1.24)	0.49	1.16 (0.95–1.42)	0.13	1.08 (0.92–1.25)	0.35
Smoking (current) (ref: none)	1.66 (1.32–2.08)	<0.0001	1.999 (1.73–2.31)	<0.0001	1.51 (1.36–1.68)	<0.0001
Eating speed (slow) (ref: normal)	0.65 (0.42–0.96)	0.03	0.97 (0.70–1.31)	0.86	1.05 (0.76–1.40)	0.77
Eating speed (fast) (ref: normal)	0.98 (0.78–1.22)	0.85	1.10 (0.96–1.25)	0.16	1.09 (0.99–1.20)	0.08
Snack after dinner (yes) (ref: no)	1.03 (0.77–1.36)	0.82	0.93 (0.78–1.11)	0.43	1.05 (0.93–1.18)	0.44
Skipping breakfast (yes) (ref: no)	1.12 (0.89–1.40)	0.34	1.20 (1.04–1.39)	0.02	1.006 (0.90–1.12)	0.91
Alcohol drinker (yes) (ref: no)	1.04 (0.83–1.30)	0.72	0.75 (0.65–0.87)	0.0001	0.73 (0.64–0.83)	<0.0001
Physical exercise (yes) (ref: no)	1.08 (0.83–1.40)	0.55	1.16 (0.98–1.37)	0.08	0.93 (0.81–1.07)	0.31

BMI, body mass index; LDL, low-density lipoprotein; HDL, high-density lipoprotein.

## Data Availability

The datasets of our study are available on reasonable request through corresponding author.

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
