# Peer review of "The Risk Factors for Development of Type 2 Diabetes: Panasonic Cohort Study 4"

_ijerph, 2022, doi:10.3390/ijerph19010571_

Round 1
Reviewer 1 Report
- The title “Population-based …..” is not very appropriate, by definition, as the title in current study. The Panasonic cohort was not sampled from general population but rather was a selected special working age cohort. The generalizability of the results and possible healthy worker effect should be discussed.
- The appropriateness of using “Having a self-reported history of diabetes” as a diagnosis of diabetes is questionable especially on the time frame of reporting, one month ago? A half year ago? Or ……? What if one with a history of diabetes but did not take any medication now?
- In the “Statistical analyses” the authors described having applied Mann-Whitney U test ( a non-parametric test comparing medians rather than means) and later said “All continuous variables are presented as mean±SD). Therefore, the authors have applied Student’s t-test and/or else?
- The authors described that this study was a “retrospective cohort study.” However, this study was not using BMI as the risk factor of interest and looked into the occurrence of newly developed diabetes (i.e., truly longitudinally looking into the effect of BMI on development of diabetes). Rather, they used BMI as a grouping factor only and recorded the development of diabetes in each BMI group. Therefore, in Table 1, it would be more appropriate to present the characteristics of individual BMI group.
- The presentation of Table 2 seemed redundant when the results of multivariate analyses (been adjusted for confounders) were presented.
- The results of multivariate analyses looked interesting. Some points revealed statistically significant protective effect which may be worth further discussion especially while targeting on life style modification such as “eating speed, slow” in the BMI < 22.0 kg/m2 group and alcohol drinking in 22.0 ≤ BMI < 25.0 kg/m2 and in BMI ≥ 25.0 kg/m2 groups.
- Several variables such as systolic blood pressure, lipids, and fasting sugars were treated continuously in the multivariate models and generated very small effects for each unit of change (probably sensitive to measurement variability). I would suggest to put these variables dichotomized or trichotomized into the models, and then comparing to see if the results are consistent.
Reviewer 2 Report
I hereby express my profound gratitude to the Editorial Board of International Journal of Environmental Research and Public Health for the opportunity to review this article.
The authors are providing an interesting topic on the potential risk factors for type 2 diabetes based on employee health status from Panasonic Corporation in Japan. With annual participations from employees, this dataset that ranges from 2008 to 2018 provides a large sample of 46,001 observants for research. The authors include a large variety of potential factors in the Cox proportional hazard regression model, separated by BMI subgroups. Multiple risk factors are identified, including age, weight gain, eating speed, cholesterol level etc. To this end, the research question is welcome and equally commendable.
Nevertheless, the paper suffers from some conceptual and methodological issues that tend to undermine its merit.
- Missing the important category of heritable risk factors. Literature suggests three main categories for risk factors that could contribute to the development of type 2 diabetes: genetic, environmental, and metabolic factors. The authors included some different measurements in the latter two categories, but failed to discuss the importance of the genetic aspect for this disease. If genetic markers are not available, a possible alternative proxy could be the family history of diabetes. Without the control on this aspect of records, any conclusion from the data analysis may be biased.
- In terms of environmental risk factors, it would be more informative if the regression con control for the types of occupations. Even though the data set come from a single company, since it’s a gigantic corporation, there might exist some variations of job types and therefore vastly different working environment and different levels of stress. The differences could thereafter induce different impacts on the development of type 2 diabetes. If job types are not available, controlling on different levels or types of income (monthly salary/hourly wage, full time/part time) would also work.
- The variable definitions and the conceptual design of the Cox regression model is a bit confusing. In terms of the weight gain variable, the manuscript does not indicate the definition, so here I assume this is the average annual change in BMI. If this is true, then the authors have employed a static Cox PH model, where all explanatory variables are documented at time t=0 (in this case year 2008), and the interpretation of coefficients are the impact of initial value (of BMI, age, cholesterol levels) on the final onset of type 2 diabetes. Then naturally, change of BMI should not serve as an explanatory variable, because the changes are only observed ex-post. However, statistically, there is an alternative Cox model that the authors can benefit from. Since the constructed panel dataset tracks annual observations for 11 years, the authors could provide Cox PH analysis with time-varying variables to answer the questions of how do varying BMI and Metabolic indicators impact the development of type 2 diabetes. For instance, the data should contain annual record of BMI, changing cholesterol level, changing behavior of smoking, changing behavior of eating speed and alcohol consumptions. This will greatly benefit the readers in understanding the dynamics of the impacts from various risk factors. Then, an interesting question would be, do BMI trends (changes in BMI in each time period) differ between diabetic vs. non-diabetic employees, and among the 3 subgroups of BMI (<= 22, 22-25, >=25).
Reviewer 3 Report
Comments:
The article “The risk factors for development of type 2 diabetes: Population-based Panasonic cohort study 4” submitted by Saijo et. al. has evaluated the risk factors contributing to the development of type 2 diabetes according to body weight mass (BMI) categories in large cohort and long period in Japanese population. This study this study, for the first time, looked into the risk factors for type 2 diabetes in Japanese population according to BMI categories. In my opinion, some revisions are needed in order to better clarify some points and avoid confusing the reader about the meaning of results achieved.
Major and minor issues are listed below:
- Introduction:
-What does it mean cohort study 4? Do you have more study with this dataset? What is their summary?
-more information needed to clarify the disease etiology.
-add some examples/explanations how the BMI could contribute to disease progression.
-what are the factors (e.g., pollution, food habit, life style, genetic), which casing higher BMI in Japanese population?
-does it represent all Japanese population or any specific region? This study is Osaka based though
- Methodology
-what is the potential bias for this study?
-is there any data availability for this study? If so, provide the link.
- Result:
-Do you think, discussing only BMI from the dataset is sufficient to draw the conclusion about diabetes development of Osaka-based Japanese population?
-What is the contribution of other test parameter like BP, cholesterol, TG etc.?
-I would recommend to add more results link with the diabetes disease development in Japanese population.
- I also encourage authors to add one more paragraph to discuss about future recommendation of the limitations list.
- What will be the next steps of this project? What will be the possible future studies for validation (e.g., collecting blood samples, study some biomarker/pathways). Any follow up studies? Saying “future research directions may also be highlighted” is not sufficient.
Round 2
Reviewer 2 Report
Thanks for the comments on the data set.
Line 189-190, "result suggests that weight gain and metabolic disorder in patients within normal weight range can raise the possibility of developing diabetes." I would assume the change in BMI is removed from the Cox analysis since I do not see the variable name any more. How do we get conclusion that weight gain will raise the possibility of diabetes onset?
Line 143-148. "The change in BMI was higher in population without developing diabetes than that with developing diabetes in population with BMI..." The wording is quite confusing here, the authors may need to rephrase their findings. -- Let's define BMI<22 as group A, BMI from 22 to 25 as group B, BMI >25 as group C. Are you saying that weight-gaining speed is (i) negatively correlated with the onset rate of diabetes in group A; (ii) but positively correlated with the development of diabetes in group B and C? This result is counter-intuitive. Although the change of BMI is not a good candidate for Cox regression, it is possible to adopt a logistic regression model to demonstrate the effects of weight-gaining speed on the development of diabetes.
